

# X-Ray: A Sequential 3D Representation For Generation

Tao Hu [1]    Wenhang Ge [2*]    Yuyang Zhao [1*]    Gim Hee Lee [1]

[1] Department of Computer Science, National University of Singapore
[2] Hong Kong University of Science and Technology (Guangzhou)

taohu@nus.edu.sg        gimhee.lee@nus.edu.sg

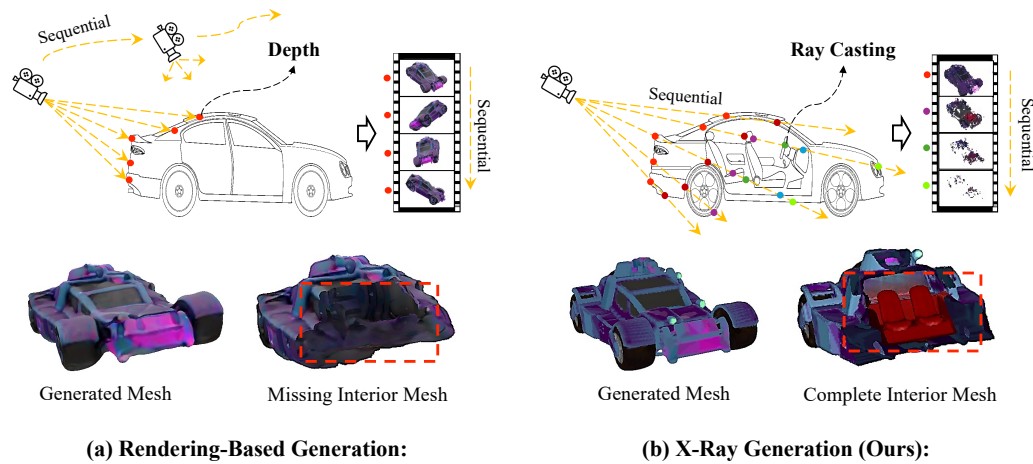

Figure 1: Comparison between the rendering-based 3D generation [49, 14] and our proposed X-Ray generation. The competitors focus on the visible outer surface in multiple camera views. In contrast, our model can sense both the visible and hidden surface in single camera view and generate the outer and inner surfaces of objects. An example of missing mesh interior from rendering-based 3D synthesis *vs.* complete mesh interior from our X-Ray generator are shown in the bottom row.

## Abstract

We introduce X-Ray, a novel 3D sequential representation inspired by the penetrability of x-ray scans. X-Ray transforms a 3D object into a series of surface frames at different layers, making it suitable for generating 3D models from images. Our method utilizes ray casting from the camera center to capture geometric and textured details, including depth, normal, and color, across all intersected surfaces. This process efficiently condenses the whole 3D object into a multi-frame video format, motivating the utilization of a network architecture similar to those in video diffusion models. This design ensures an efficient 3D representation by focusing solely on surface information. Also, we propose a two-stage pipeline to generate 3D objects from X-Ray Diffusion Model and Upsampler. We

---

[*]Co-second authors.

38th Conference on Neural Information Processing Systems (NeurIPS 2024).

demonstrate the practicality and adaptability of our X-Ray representation by synthesizing the complete visible and hidden surfaces of a 3D object from a single input image. Experimental results reveal the state-of-the-art superiority of our representation in enhancing the accuracy of 3D generation, paving the way for new 3D representation research and practical applications. Our project page is in https://tau-yihouxiang.github.io/projects/X-Ray/X-Ray.html.

# 1  Introduction

General, accurate, and efficient 3D representations are three of the most critical requirements for 3D generation [23, 29, 32, 21]. The significance of this goal stems from the ever-expanding array of applications reliant on 3D technology, ranging from virtual reality and augmented reality to computer-aided design and beyond. Previous approaches to 3D representation such as meshes, point clouds, voxels, Neural Radiance Fields (NeRF) [36, 45, 58, 24, 15] and 3D Gaussian Splatting [21] possess unique strengths respectively, but face challenges in concurrently satisfying the three requirements for 3D synthesis. Specifically, meshes are widely used in 3D modeling, while they are constrained by their topology when describing complex objects, which limits their generative capacity. Point clouds offer a more flexible capture of the object geometries but lack continuous and dense feature extraction [12, 42]. Voxels simplify spatial reasoning at the cost of significant rising memory complexity with increasing resolution. Neural representations, such as NeRF [36] and 3D Gaussian Splatting [21], offer an impressive leap in rendering photorealistic scenes. Nevertheless, the 3D object are predicted by multi-view images with a relatively long optimized period.

Recently, rendering-based 3D generative methods [26, 57, 48, 14, 53, 49, 18] have gained significant attention for their ability to achieve general and even efficient 3D generation by incorporating neural representations [36, 21] with Transformers [52] or Diffusion Models [43, 2]. However, these methods have a critical limitation: they cannot completely generate objects that include both visible and hidden surfaces. This limitation arises from the methodological design of rendering-based 3D generation, which relies on the 2D supervision of rendered images. As a result, these methods primarily focus on reconstructing the visible external surfaces of objects, while neglecting the internal hidden surfaces. This oversight leads to incomplete or unrealistic reconstructions, as shown in Fig. 1 (a).

In this paper, we propose the X-Ray representation to overcome this limitation of incomplete generation while maintaining the efficiency and generalization required for 3D generation. As illustrated in Fig. 1 (b), our X-Ray, inspired by x-ray imaging in the medical field, can see through the entire object. It efficiently captures and stores information about both visible and hidden surfaces. Consequently, the hidden interior of the object can be fully reconstructed.

Our X-Ray is designed to capture the shape (depth and normal) and appearance attributes (color) along all the sequentially intersected surfaces through ray casting. We transpose the collected slim grid voxels into a multi-frame surface representation, which significantly reduces the data footprint while preserving essential detail. Moreover, the compatibility of our X-Ray's data structure with sequential 3D representation in video formats, as illustrated in Fig. 2, opens novel pathways for leveraging video diffusion models in 3D generation. Specifically, by treating our X-Ray representation as sequences of frames, we first harness the power of the video diffusion model [2] and then utilize the video upsampler [7] to generate 3D objects from low to high resolution. As a result, our approach yields high-quality results while inheriting the advanced capabilities and efficiency of video processing.

We demonstrate the advantages of our X-Ray representation through comprehensive experiments, showcasing its superiority, especially completeness in 3D object generation. We train and evaluate our method in image-to-3D reconstruction task and pure 3D generation task. The experimental results reveal that the proposed X-Ray achieves a significant leap forward in the quality of 3D object generation, positioning it as a feasible solution to longstanding challenges in the field. The main contribution of the paper can be summarized as follows:

1. We present X-Ray, a novel 3D representation that encode the whole visible and hidden surfaces in to video format through ray casting algorithm for maintaining generalization, accuracy and efficiency.

2. We propose the generative model of our X-Ray via video diffusion model and video upsampling model, enabling low-to-high generation of 3D objects from single images.

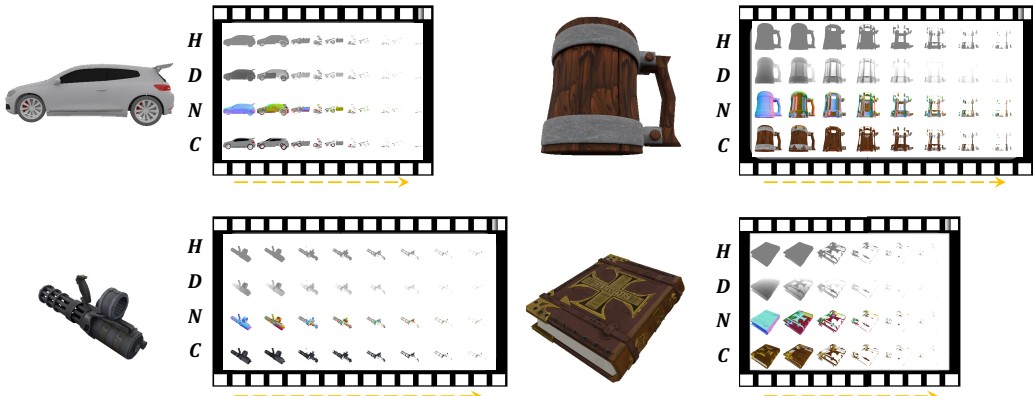

Figure 2: Samples of our X-Ray 3D sequential representation. Given a viewpoint, we capture the 3D attributes multi-layer surface frames, including hit **H**, depth **D**, normal **N**, and color **C**, in a video format. Noted that the number of frames in an X-Ray varies depending on the complexity of the 3D objects. The dotted yellow lines indicate the ray or sequence direction.

3. We showcase the state-of-the-art performance of our X-Ray in 3D generation quality, setting a new benchmark for Image-to-3D modeling.

## 2 Related Work

### 2.1 Representation for 3D Models

Handling 3D data is much more complex and resource-intensive than dealing with 1D (*e.g.* text and voice) and 2D (*e.g.* image) data. This complexity makes it imperative to find effective ways to organize, process, and infer 3D information. Traditional methods for representing 3D data include meshes, which are good for creating detailed visuals but hard to be generalized; point clouds, which are simple and useful for capturing real-world scenes but lack consistent and dense structure in 3D creation; Although, 3D Gaussian Splatting [21] smooths point cloud data into continuous surface but requires additional an initial point cloud as shape, making them less flexible for 3D synthesis; voxels which are excellent for detailed volumetric data but require much computing resources. Multi-Plane Images [35, 51] try to extent the depth concept to multi-layer with a fixed distance, but they can only describe the visible surface toward camera.

Recent advancements in 3D representation have primarily focused on point-level details and implicit functions, such as Occupancy [32], Signed Distance Fields (SDF) [55, 11], Triplanes [10, 16], and Neural representations [36, 21]. These methods have significantly enhanced modeling and rendering capabilities. Occupancy models map the location of any 3D point to its probability of being inside or outside an object, offering a probabilistic approach to shape definition. SDFs [55, 19] refine this concept by quantifying the nearest signed surface distance from any given point, improving the precision of surface representations. Triplanes [3] employ intersecting 2D planes to provide a more efficient route to 3D representation, albeit with some detail loss. NeRFs and Gaussian Splatting [21, 36] produce remarkably realistic renderings from a limited number of viewpoints but require extensive computational effort. Despite these advancements, implicit function-based models often face challenges in extracting full and high-resolution 3D features, hindering high-quality generalization. Empirically, representations that focus on surface is more efficient, and representations with grid representation are more easily to be generalized [37]. Consequently, capturing all surface attributes and organizing in a dense but lightweight data structure renders our X-Ray an accurate, efficient and generalized representation. Noted that our X-Ray is similar to Depth peeling method [9], which is designed for rendering transparent surfaces, while X-Ray transform any 3D object in video format. Besides, our main contribution is using video diffusion as generator to generate objects.

## 2.2 Generative Models for 3D Generation

Recent 3D generative models can be primarily categorized into two types: diffusion-based [30, 37, 38] and rendering-based [14, 13, 49, 41, 34, 54, 33, 47]. Diffusion-based models fall under direct 3D supervision, while rendering-based models belong to indirect 2D image supervision. Diffusion-based generative models have emerged as powerful tools for 3D generation, leveraging stochastic diffusion processes to gradually transition from noise to structured 3D objects. These models, such as DPM [30], DiT-3D [37], and Point-E [38], have demonstrated remarkable ability in generating high-quality 3D point clouds. They operate by iteratively refining a random noise distribution into a coherent structure that resembles the target 3D shape, capturing complex geometries and surface details with high fidelity. The strength of these models lies in their capacity to model the distribution of 3D points in a continuous space, allowing for the generation of 3D objects with nuanced variations and detailed textures. However, both point-based networks [28, 37, 17] and voxel-based networks [37] is limited by generating high-resolution objects. Another group of methods [41, 34, 54, 33, 47] adopt Score Distillation Sampling (SDS) as prior to train a NeRF [36] or 3D Gaussian Splatting [21] for 3D generation. However, it is not efficient for optimizing a number of different views over a short period.

On the other hand, rendering-based generative models focus on the visible aspect of 3D generation, transforming abstract 3D representations into detailed and photorealistic images or videos. Models such as LRM [14], Open-LRM [13], LGM [48], DMV3D [57], and TripoSR [49] employ advanced rendering techniques to achieve this. However, rendering-based models are optimized only for the visible surfaces of objects, making it difficult to synthesize the invisible or internal surfaces.

In response to these challenges, our approach utilizes a video diffusion model as the foundation for developing 3D X-Ray. This strategy benefits from the strengths of existing video diffusion models while innovatively addressing the limitations of rendering-based techniques, offering a more comprehensive solution for 3D generation that is sensitive to both visible and hidden parts of objects.

# 3 Our X-Ray Representation

In this section, we detail our X-Ray representation, which includes both encoding and decoding process to facilitate the conversion between 3D mesh formats and our X-Ray representation. Specifically, the encoding process converts a 3D mesh into our proposed X-Ray format, and the decoding process converts our X-Ray back into a 3D mesh.

## 3.1 Encoding

Given a 3D object under an arbitrary camera view, we apply the ray casting algorithm to encode a 3D object mesh into the proposed X-Ray representation. The ray casting algorithm plays a crucial role in both computer graphics and computational geometry, where it is used for scene rendering, visibility determination, and addressing geometric queries. Specifically, a ray is emitted from camera center into the environment and all the interactions of this ray with target 3D objects are captured sequentially. For each ray $r \in \mathcal{R}$ within the field of view that intersects with $L$ sequential faces in the mesh, we record their 3D attributes which include depth (distance to camera center) $\mathbf{D}_r = (\mathbf{d}_1, \mathbf{d}_2, ..., \mathbf{d}_L) \in \mathbb{R}^{L \times 1}$, normal $\mathbf{N}_r = (\mathbf{n}_1, \mathbf{n}_2, ..., \mathbf{n}_L) \in \mathbb{R}^{L \times 3}$, and color $\mathbf{C}_r = (\mathbf{c}_1, \mathbf{c}_2, ..., \mathbf{c}_L) \in \mathbb{R}^{L \times 3}$. To indicate surface presence, we denote Hit $\mathbf{H}_r = (\mathbf{h}_1, \mathbf{h}_2, ..., \mathbf{h}_L) \in \mathbb{R}^{L \times 1}$ to indicate whether there is a surface. Since $L$ is usually very small (Sec. 5.3), we denote the efficient and thin grid voxels $\mathbf{X} \in \mathbb{R}^{H \times W \times L \times 8}$ as the object representation, where the ray $\mathbf{X}_{ij}$ with image coordinate $[i, j]$ can be represented as:

$$\mathbf{X}_{ij} = \mathbf{X}_r = (\mathbf{H}_r, \mathbf{D}_r, \mathbf{N}_r, \mathbf{C}_r) \in \mathbb{R}^{L \times 8}. \tag{1}$$

Note that $\mathbf{X}[i, j, k] = 0$ when there is no surface for the $k^{th}$ layer at the image ray coordinate $[i, j]$. Examples of X-Ray encoding are shown in Fig. 2. Through the encoding process, we can transform any mesh into a sequential representation with varying lengths, same as a video with different numbers of frames. Finally, we transpose the voxels as **X-Ray** $\mathbf{X} \in \mathbb{R}^{L \times 8 \times H \times W}$, resembling a video format with $L$ frames, each with a resolution of $H \times W$ and 8 feature channels.

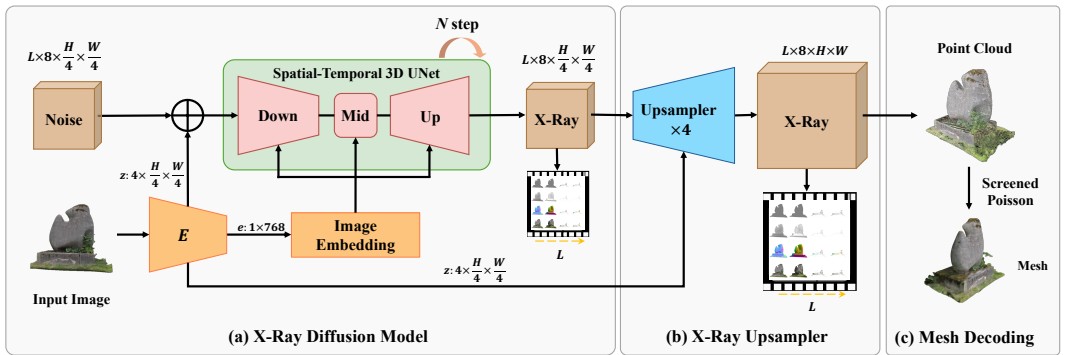

Figure 3: Overview of our proposed generative pipeline for the X-Ray 3D representation. There are three main components: (a) The X-Ray diffusion model, which generates a low-resolution X-Ray from an image input. (b) The upsampler, which enlarges the low-resolution X-Ray into $4\times$ high resolution. (c) The mesh decoding model, which decodes the high-resolution X-Ray into a point cloud with color and normal, and then converts it into the final generated mesh.

## 3.2 Decoding

The decoding process converts the X-Ray representation back into a 3D mesh. To achieve this, we first convert the video format of X-Ray into a point cloud and subsequently apply the Screened Poisson algorithm [20] to transform the point cloud into a 3D mesh.

**X-Ray $\rightarrow$ Point Cloud.** Given an X-Ray, we first compute the 3D object's point cloud $\mathbf{P_r} = \{\mathbf{P_x}, \mathbf{P_n}, \mathbf{P_c}\}$ for Ray $r$, including location $\mathbf{P_x}$, color $\mathbf{P_c}$, and normal $\mathbf{P_n}$ defined by the equation:

$$\mathbf{P_x} = \mathbf{r}_o + \mathbf{D}_r \cdot \mathbf{r}_d, \quad \mathbf{P_n} = \mathbf{N}_r, \quad \mathbf{P_c} = \mathbf{C}_r, \quad \text{when} \quad \mathbf{H}_r = 1. \tag{2}$$

$\mathbf{r}_o$ and $\mathbf{r}_d$ denote the origin and direction of the camera ray, respectively. Furthermore, $\mathbf{D}_r, \mathbf{N}_r, \mathbf{C}_r$ and $\mathbf{H}_r$ are the depth, surface normal, color and hit attributes of the ray defined in Eq. 1. Upon processing all camera rays, we obtain a comprehensive point cloud $\mathbf{P} = \{\mathbf{P}_r\}_{r \in \mathcal{R}}$ representation that includes location, normal, and color attributes of the 3D object.

**Point Cloud $\rightarrow$ Mesh.** The Screened Poisson algorithm [20] for converting point clouds with color and normal into 3D colored meshes is a classic method that leverages the mathematical principles of the Poisson equation. The core idea involves solving a variation of the Poisson equation to interpolate a smooth surface that fits the input point cloud. The Poisson equation is a partial differential equation of the form: $\nabla^2 \phi = f$, where $\nabla^2$ denotes the Laplace operator (which represents the divergence of the gradient of a function), $\phi$ is the potential field to be solved, and $f$ is a scalar function representing the divergence of the vector field derived from the input point cloud. In the context of point cloud to 3D mesh conversion, the algorithm first employs the given normal to define a vector field that suggests the orientation of the surface at each point. The divergence of this vector field serves as the function $f$ the Poisson equation.

**Encoding-Decoding Intrinsic Error.** The encoding-decoding process will introduce a intrinsic and minor reconstruction error that varies with the number of layer $L$ and the frame resolution $(H, W)$. To explore this, we conduct an experiment in Sec. 5.3 aimed at analyzing these variables to identify their optimal values. Our goal is to achieve a balance where all pertinent information is preserved while maintaining a lightweight model.

## 4 X-Ray for 3D Generation

Our primary objective of introducing a new 3D representation model is to facilitate the generation of 3D objects from single images. The challenge lies in accurately predicting the characteristics that are not immediately visible on the first surface when only a single image is available. To overcome this challenge, we utilize diffusion and upsampling models for X-Ray synthesis. Given that our proposed X-Ray is in video format, we leverage advanced Video Diffusion models as our backbone. To exploit this structure for high-resolution X-Ray synthesis, we incorporate principles from advanced video

diffusion models as our foundational framework. Notable models in this domain include Stable Video Diffusion (SVD) [2], VideoFusion [31], and the state-of-the-art Sora. To efficiently train the diffusion model, we begin by training a low-resolution X-Ray diffusion model that generates X-Ray from a single image. Subsequently, we employ an upsampler to enhance these synthesized X-Rays to high resolution. This two-step approach ensures a more manageable and efficient training process, gradually improving the quality of the output.

**Framework Overview.** Fig. 3 presents an overview of our generative model using X-Ray representation (*cf.* Sec. 3.1) for 3D generation. Our X-Ray diffusion model (*cf.* Sec. 4.1) operates at the core of the framework, transforming random Gaussian noise into a low-resolution X-Ray representation conditioned by an input image. These low-resolution X-Rays are then enhanced to high resolution through the application of a 3D Spatial-Temporal Upsampler (*cf.* Sec. 4.2). Finally, the high-resolution X-Rays are decoded into 3D meshes using a combination of point cloud transformation and the Screened Poisson algorithm (*cf.* Sec. 3.2).

## 4.1 X-Ray Diffusion Model

**Diffusion models** [43] are generative models that transform a random noise distribution into a data distribution through a reverse process, counteracting a forward process that incrementally adds Gaussian noise to the data. The forward process is a Markov chain described by $x_t = \sqrt{\alpha_t}x_{t-1} + \sqrt{1 - \alpha_t}\epsilon$, where $x_t$ represents the data at step $t$, $\alpha_t$ controls the noise level, and $\epsilon \sim \mathcal{N}(0, I)$ is sampled noise. The reverse process, aimed at reconstructing the original data from noise, is modeled by a neural network predicting the noise added at each step or directly denoising the data, following $x_{t-1} = \frac{1}{\sqrt{\alpha_t}} \left( x_t - \frac{1-\alpha_t}{\sqrt{1-\alpha_t^2}}\epsilon_\theta(x_t, t) \right)$, with $\epsilon_\theta(x_t, t)$ being the predicted noise. Training involves optimizing the network to minimize the difference between the original and reconstructed data, effectively learning to invert the noise addition process given by:

$$\mathcal{L}_{dm} = \mathbb{E}_{x,\epsilon\sim\mathcal{N}(0,1),t} \left[ \|\epsilon - \theta(x_t, t)\|^2 \right], \tag{3}$$

where $t$ is uniformly sampled from the set $\{1, \ldots, T\}$.

**Diffusion Model for X-Ray.** A prevalent technique in diffusion models is the utilization of latent spaces with a VQ-VAE [7] to perform the initial data transformation to compress the data. This method poses a significant challenge for our X-Ray representation as it requires the development of a VQ-VAE model from scratch due to the absence of a suitable off-the-shelf latent model for X-Ray, which would consequently increase our training burden. Another promising approach for efficiently training high-resolution generators is the cascaded synthesis pipeline. This method, exemplified by works such as Imagen [44], DeepFloyd IF [1], and Stable Cascaded [40], involves progressively training the diffusion model or upsampling network from lower to higher resolutions. Given our limited computing resources, we opted to implement this cascaded upsampling strategy. This technique facilitates a more gradual and controlled enhancement of X-Ray quality, providing a more flexible and efficient alternative to traditional latent space diffusion models.

Specifically, we use the Spatial-Temporal 3D U-Net network from Stable Video Diffusion [2] for our diffusion model to generate low-resolution X-Rays, with modifications to the input and output channels. As shown in Fig. 3, the input image is sent to the encoder $\mathbf{E}$, producing an image latent $z$ via image VAE [7] and an embedding $e$ via a ViT [5]. $z$ is concatenated with the X-Ray latent as input, and $e$ interacts with the 3D U-Net through cross attention [2] to finally output the denoised latent. This model employs spatial-temporal attention mechanisms to alternately extract features from 2D frame spaces and 1D surface layer sequences, enhancing its ability to process and interpret the different layers of the X-Ray. This approach allows for nuanced handling of the temporal information inherent in sequential X-Ray data, crucial for achieving high-quality diffusion results.

## 4.2 X-Ray Upsampler

The X-Ray upsampler focuses on enhancing previously generated X-Rays to a higher resolution. We considered two potential methods: point cloud up-sampling and video up-sampling. Encoding low-resolution X-Rays into a point cloud with color and normal information is straightforward (see Sec 3.1). However, point cloud up-sampling often increases only the number of points without effectively enhancing attributes like texture and color due to its unstructured nature. To improve

efficiency and consistency, we adopted a video up-sampling approach using a spatial-temporal VAE decoder from Stable Video Diffusion (SVD) [2]. We concatenate the previous image latent $z$ with the low-resolution X-Ray as input and output the high-resolution X-Ray. The model upsamples previously synthesized low-resolution X-Ray frames fourfold while preserving the original layer number $L$. It applies attention at both the 2D surface frame and 1D surface layer levels, enhancing frame resolution and overall quality. This makes the X-Ray diffusion model, followed by the upsampler, a more integrated and effective solution.

**Loss**. The loss function for the Upsampler differs notably from that of the diffusion model. While the diffusion model loss typically addresses volumetric or textural aspects, the Upsampler loss concentrates specifically on the surface area accuracy, reflecting the critical importance of maintaining high fidelity in the enhanced images. The loss function we use for the Upsampler is given by:

$$\mathcal{L}_{up} = \|\mathbf{X}_{gt}[\mathbf{H}_{gt}] - \mathbf{X}_{up}[\mathbf{H}_{gt}]\|^2 + \|\mathbf{H}_{gt} - \mathbf{H}_{up}\|^2, \tag{4}$$

where $\mathbf{X}_{gt}[\mathbf{H}_{gt}]$ represents the ground-truth high-resolution X-Ray at hit surface and $\mathbf{X}_{up}[\mathbf{H}_{gt}]$ denotes the Upsampler's output at hit surface, and $\mathbf{H}_{gt}$, $\mathbf{H}_{up}$ denotes the ground-truth and upsampled Hit, respectively. The loss is computed as the squared Euclidean distance between these two matrices, quantifying the pixel-wise discrepancy in surface details. This metric effectively ensures that the upsampling process preserves essential surface features, thereby optimizing the quality and utility of the resulting high-resolution X-Ray.

## 5 Experiments

### 5.1 Dataset and Implementation

**Datasets.** We train our X-Ray pipeline using a subset of the Objaverse dataset [4], which have been removed entries with missing textures and inadequate prompts as outlined in [48]. This subset consists of more than 60,000 3D objects. For each object, we select 8 random camera views, covering azimuth angles from -180 to 180 degrees and elevation angles from 0 to 45 degrees with camera distance to object center fixed at 1.2. The images are then rendered using Blender Software, and the corresponding X-Rays are generated through the ray casting algorithm provided by the trimesh library [50]. Through these processes, we create a dataset of approximately 480,000 paired images and X-Rays to train the generative model. For the evaluation datasets, we adopt two commonly adopted datasets: Google Scanned Objects [6] and OmniObject3D [56], to assess generative performance via single-view reconstruction tasks.

**Metrics.** Recent 3D generative models [26, 18, 27, 14, 49] lack unified reconstruction evaluation metrics due to the challenge of determining an object's size and orientation from a single image [46]. To ensure fair comparisons with state-of-the-art methods, we align all methods before evaluation. The predicted and ground-truth 3D objects will be normalized to a range of -0.5 to 0.5 along all three axes and face forward the same $-z$ axis. We then align them using the Iterative Closest Points (ICP) algorithm and calculate Chamfer Distance (CD) in $L1$ norm and F-Score (FS) at threshold 0.1 (FS@0.1) for reconstruction.

**Implementation Details.** Our X-Ray diffusion model is based on the Spatrial-Temporal 3D UNet architecture used in Stable Video Diffusion (SVD) [2], modified to synthesize 8 channels: 1 hit channel, 1 depth channel, 3 color channels, and 3 normal channels, compared to the original 4 channels. During training, we maintain a learning rate of 0.0001 using the AdamW optimizer. Since different X-Rays have varying numbers of layers, we pad or truncate them to a uniform 8 layers for efficient batching and training. Each layer's frame has dimensions of $64 \times 64$. For the upsampler, each layer's output remains at 8 channels, but the resolution of each frame is increased to $256 \times 256$ to enhance detail and clarity in the upscaled X-Ray. The entire training pipeline is conducted on 8 NVIDIA A100 GPU servers for two weeks. During inference, the 3D generation process takes approximately 7 seconds: about 1 second for the diffusion model, 1 second for the upsampler, and 5 seconds for mesh decoding. As for GPU usage during inference, the GPU memory required is 4.8 GB for X-Ray diffusion model and 2.5 GB for X-Ray Upsampler.

Table 1: Comparison with other 3D representations in Efficiency.

| Metric | 3D Grid | Multi-View Depths (8 views) | MPI (8 planes) | Point Cloud (200,000 points) | X-Ray (8 layers) |
|---|---|---|---|---|---|
| Memory (↓) | 67.09 MB | 1.57 MB | 1.57 MB | **0.90 MB** | **0.62 MB** |
| Encoding Method | Voxlization | Rendering | Slicing & Rendering | Sampling | Ray Casting |
| Encoding Time (↓) | 0.105 s | 0.045 s | 0.049 s | **0.013 s** | **0.016 s** |
| Decoding Method | Poisson | Fusion & Poisson | Poisson | Poisson | Poisson |
| Decoding Time (↓) | ∼5 s | ∼10 s | ∼5 s | ∼5 s | ∼5 s |
| CD(↓) | 7.7e-3 | 1.1e-2 | 8.9e-3 | **7.2e-3** | 7.8e-03 |

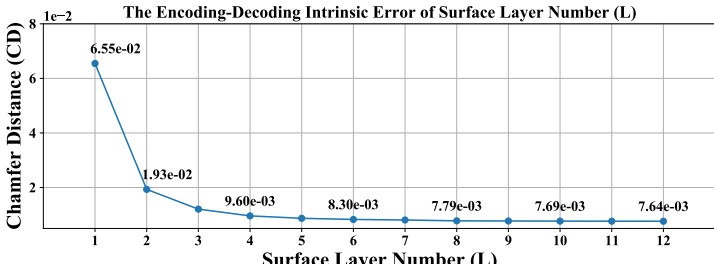

(a) The intrinsic error of layer number ($L$), when $H = W = 256$

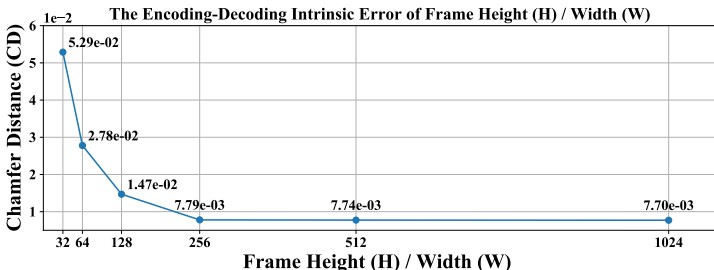

(b) The intrinsic error of frame height ($H$) or width ($W$) when $L = 8$

Figure 4: The encoding-decoding intrinsic error of different frame resolutions and number of layers.

## 5.2 Efficient Comparison with Different 3D Representation

We compared the efficiency of different representations using 500 3D meshes from random selected models. The results showed that both point cloud and X-Ray were highly efficient, with lower memory, faster encoding & decoding times. However, the X-Ray had the advantage of being reorganizable as a video format for diffusion models, leading to better performance.

## 5.3 Analysis of Encoding-Decoding Intrinsic Error

Due to the finite number of layer $L$ and resolution $H, W$ of X-Ray, a slight intrinsic error is inevitable during the encoding of 3D meshes into X-Ray format and the subsequent decoding back into 3D meshes. To quantitatively assess this error, we conducted an experiment to evaluate intrinsic error via Chamfer Distance (CD) ($L1$ norm) between the original (ground-truth) mesh and the encoding-decoding mesh across various resolutions and layers. In the experiments, we set the frame resolution through a series of predefined values: 32, 64, 128, 256, 512, and $1,024$ and vary the number of layers from 1 to 12. Fig. 4a indicates that the intrinsic error decreases as the number of layers in our X-Ray representation increases, becoming convergent after 8 layers. Similarity as illustrated in Fig. 4b, the intrinsic error decreases with increasing resolution and stabilizes after 256. Therefore, for a balance between accuracy and efficiency, we use a resolution $8 \times 256 \times 256$ with $H = W = 256$ and $L = 8$ in our experiments. Compared with the dense volume achieving a $256 \times 256 \times 256$ resolution for voxel-based methods, our X-Ray representation is significantly efficient for focusing only on surfaces and reducing the data volume by 96.88%.

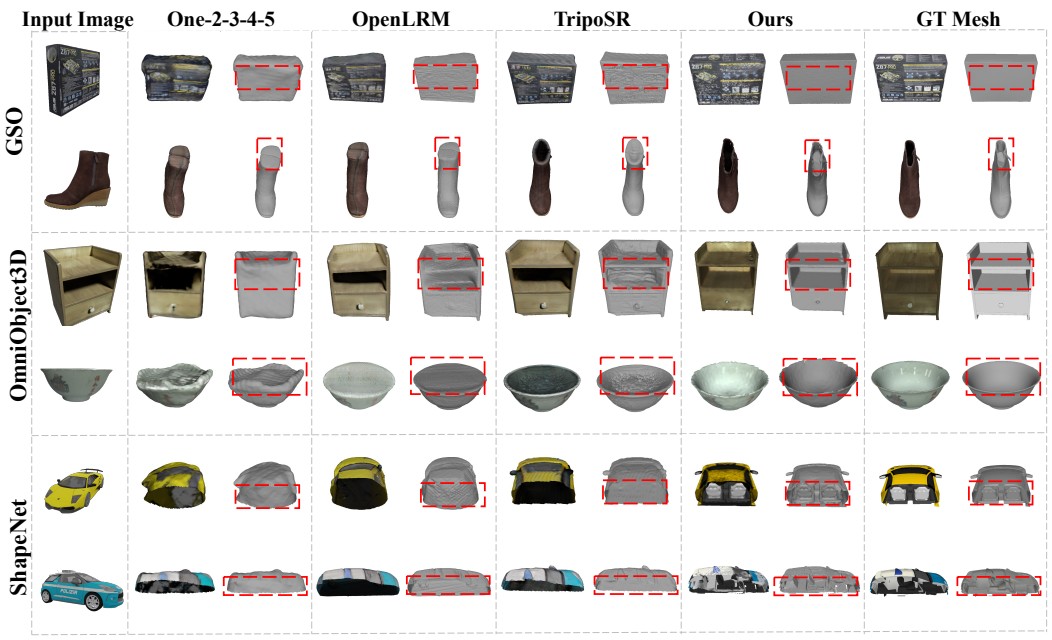

Figure 5: Quantitative Comparison in Image-to-3D Generation.

## 5.4 Quantitative Comparison

For image-to-3D mesh generation on the GSO [6] and OmniObject3D [56] datasets, we build a new benchmark that randomly selects 500 image-mesh pairs from each dataset. We re-ran the baselines using their released source code and used normalized Chamfer distance and F-score for a fair comparison. The results, summarized in Tab. 2, show that our X-Ray method achieves significant improvements over all previous rendering-based state-of-the-art methods [25, 18, 8, 13, 49] on both datasets, with a relative **33%** improvement ($0.084 \rightarrow 0.056$) in Chamfer distance and also a significant improvement in FS@0.1 ($0.878 \rightarrow 0.973$, where the maximum is 1) on GSO dataset and similar performance on OmniObject3D dataset. This demonstrates the superiority of our approach over rendering-based methods.

Table 2: Quantitative reconstruction comparison on the GSO [6] and OmniObject3D [56] datasets

| Datasets | Metrics | One-2-3-45 [25] | ZeroShape [18] | TGS [8] | OpenLRM [13] | TripoSR [49] | X-Ray (Ours) |
|---|---|---|---|---|---|---|---|
| GSO [6] | CD ↓ | 0.175 | 0.136 | 0.096 | 0.143 | 0.084 | **0.056** |
| | FS@0.1 ↑ | 0.465 | 0.627 | 0.803 | 0.621 | 0.878 | **0.973** |
| OmniObject3D [56] | CD ↓ | 0.187 | 0.138 | 0.091 | 0.148 | 0.080 | **0.054** |
| | FS@0.1 ↑ | 0.490 | 0.619 | 0.822 | 0.664 | 0.892 | **0.972** |

## 5.5 Qualitative Comparison

A qualitative comparison effectively demonstrates the advantages of our proposed X-Ray generation method. Using the three datasets mentioned, we selected single images as input and generated 3D meshes without and with textures, as shown in Fig. 5. Our proposed method has three key advantages: 1. It can decompose shape and appearance to accurately reconstruct flat surfaces (Rows 1 and 3); 2. It can detect the sealing of containers (Rows 2 and 4); 3. It can generate the internal structure of objects within their outer surfaces (Rows 5 and 6). These obvious advantages highlight the our effectiveness.

## 5.6 Failure Cases.

The failure cases highlight the limitations of the current generative model based on X-Ray representation when the number of frame layers is very large. As illustrated in Fig. 6, given an input image containing a complex object, such as an exquisite hamburger, the number of frame layers $L$ of the encoded or generated X-Ray tends to exceed the maximum length of 8. Consequently, any surface

behind layer 8 will be omitted, resulting in missing parts of the reconstructed mesh. The solution is to increase the value of $L$ so that the X-Ray can represent more surfaces. However, this would also increase computing resource requirements. We will reconsider the sparsity of deeper surface frames and propose a more efficient generative model to overcome these failure cases.

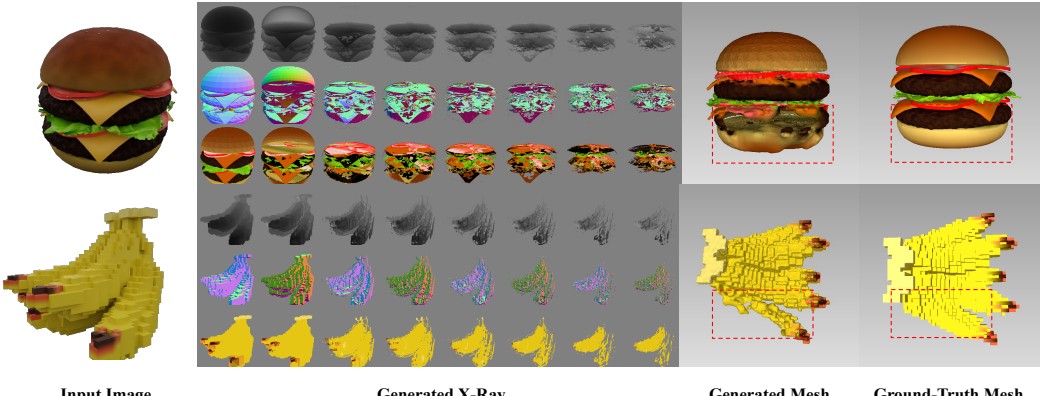

| Input Image | Generated X-Ray | Generated Mesh | Ground-Truth Mesh |

Figure 6: Failure cases. The generated meshes will miss behind parts because of the limited number of frame layers.

# 6    Conclusion

In this work, we introduced a novel X-Ray representation for 3D objects that encompasses both visible and hidden surfaces within the camera's field of view, unlike recent rendering-based methods that typically focus only on the visible surface. We demonstrated the effectiveness of the X-Ray approach in single-view 3D generation tasks. Our generative model shows that the underlying generator for X-Ray shares foundational similarities with existing video diffusion models, allowing us to leverage their advantages. Experimental results highlight the outstanding performance of our method.

**Limitations.** Our generator uses the Stable Video Diffusion (SVD) pipeline to produce high-quality X-Ray. However, the X-Rays generated consist of an uncertain number of sequential layers, with the posterior layers tending to be more and more sparse, which can be redundant. Additionally, the generated mesh lacks enough smoothness and has missing part when X-Ray is truncated. We plan to explore advanced network architectures, such as Large Language Model, that can better handle the complexities of X-Ray data, including layer sparsity and sequential format. Additionally, we aim to investigate further applications of the X-Ray representation to broaden its utility and impact in 3D modeling and beyond.

# Acknowledgement

This research / project is supported by the National Research Foundation (NRF) Singapore, under its NRF-Investigatorship Programme (Award ID. NRF-NRFI09-0008).

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

# A   Appendix

The Appendix section contains detailed information on the network structure (including the X-Ray diffusion model and upsampler), ablation studies, additional experimental quantitative and qualitative results, failure cases, and key source code.

## A.1   Network Details

**X-Ray Diffusion Model** As outlined in our study, we utilized the 3D Spatial-Temporal UNet as the foundational architecture for our diffusion model. For image-to-3D generation, the input includes 4-channel image latent and 8-channel noise X-Ray and the output has 8 channel to denoise X-Ray. Initially, we loaded all network parameters and fine-tuned the model on our training dataset. Despite these efforts, the final performance fell short of expectations. We hypothesized that this underperformance stemmed from the limited size of our training dataset, which was insufficient for such a parameter-rich model, as well as a substantial domain gap between the original video data and our X-ray data.

To address these issues, we reduced the network size to $10\%$ of its original configuration and trained the model from scratch using a significantly larger batch size. This approach proved highly effective, as the final performance exceeded the current state-of-the-art methods by a considerable margin. Consequently, our findings suggest that a smaller, more focused network trained from scratch can be more effective than a larger pre-trained model when faced with limited and domain-specific datasets. This approach not only enhances performance but also highlights the importance of customizing the model size and training strategy to the specific characteristics of the data. Our results indicate that careful consideration of the network architecture and training regimen is crucial for optimizing performance, particularly in specialized applications such as X-ray generation. This study provides valuable insights into model adaptation and training strategies that can be applied to other domains facing similar challenges.

The following JSON file contains the configuration details for our X-Ray Spatial-Temporal UNet.

```json
{
    "_class_name": "UNetSpatioTemporalConditionModel",
    "addition_time_embed_dim": 256,
    "block_out_channels": [
        64,
        128,
        256,
        256
    ],
    "cross_attention_dim": 1024,
    "down_block_types": [
        "CrossAttnDownBlockSpatioTemporal",
        "CrossAttnDownBlockSpatioTemporal",
        "CrossAttnDownBlockSpatioTemporal",
        "DownBlockSpatioTemporal"
    ],
    "in_channels": 12,
    "latent_channels": 8,
    "layers_per_block": 1,
    "num_attention_heads": [
        4,
        8,
        16,
        16
    ],
    "num_frames": 8,
    "out_channels": 8,
    "projection_class_embeddings_input_dim": 768,
    "sample_size": 64,
    "transformer_layers_per_block": 1,
    "up_block_types": [
        "UpBlockSpatioTemporal",
```

```
33        "CrossAttnUpBlockSpatioTemporal",
34        "CrossAttnUpBlockSpatioTemporal",
35        "CrossAttnUpBlockSpatioTemporal"
36    ]
37 }
```

## A.2  X-Ray Upsampler

The X-Ray diffusion model enables the generation of low-resolution 3D objects. To enhance resolution and improve performance, our X-Ray upsampler increases the frame resolution by a factor of 4. For image-to-3D generation, we concatenate the 4-channel image latent representation with the 8-channel low-resolution X-Ray, producing an 8-channel high-resolution X-Ray.

This process begins with the diffusion model creating a coarse, low-resolution 3D representation that captures the essential structure of the object. The upsampler then refines this representation, significantly improving the resolution and adding finer details that contribute to the realism of the 3D model. By combining the latent image features with the initial low-resolution X-Ray data, we ensure that the final high-resolution output retains the context and nuances of the original image while also incorporating the detailed structural information provided by the X-Ray data.

The following JSON file contains the configuration details for our X-Ray Upsampler.

```
1  {
2    "_class_name": "AutoencoderKLTemporalDecoder",
3    "block_out_channels": [
4      128,
5      256,
6      512
7    ],
8    "down_block_types": [
9      "DownEncoderBlock2D",
10     "DownEncoderBlock2D",
11     "DownEncoderBlock2D"
12   ],
13   "force_upcast": false,
14   "in_channels": 4,
15   "latent_channels": 12,
16   "layers_per_block": 2,
17   "out_channels": 8,
18   "sample_size": 768,
19   "scaling_factor": 1.0
20 }
```

## A.3  Ablation Studies

### The Effect of Diffusion Model

As described in Sec. A.1, we train the diffusion model with different configurations, including varying model sizes and how to conduct initialization. Specifically, we evaluate three models:

1. Finetuned Original UNet
2. Randomly Initialized Original UNet
3. Randomly Initialized UNet with 10% Parameters

The experimental evaluation results on the GSO [6] dataset are illustrated in Tab. 3. These results allow us to analyze the impact of different initialization and scaling strategies on the performance of the diffusion model. By comparing the outcomes, we can identify the trade-offs between model size, training time, and overall accuracy. We observed that finetuning the model did not introduce significant improvement because of the domain gap between video and X-Ray data. Also, it is not necessary to adopt a large diffusion model for Objeverse dataset [4] for significantly reducing the batch size. Thus, we adopt the randomly initialized UNet with 10% parameters as our diffusion

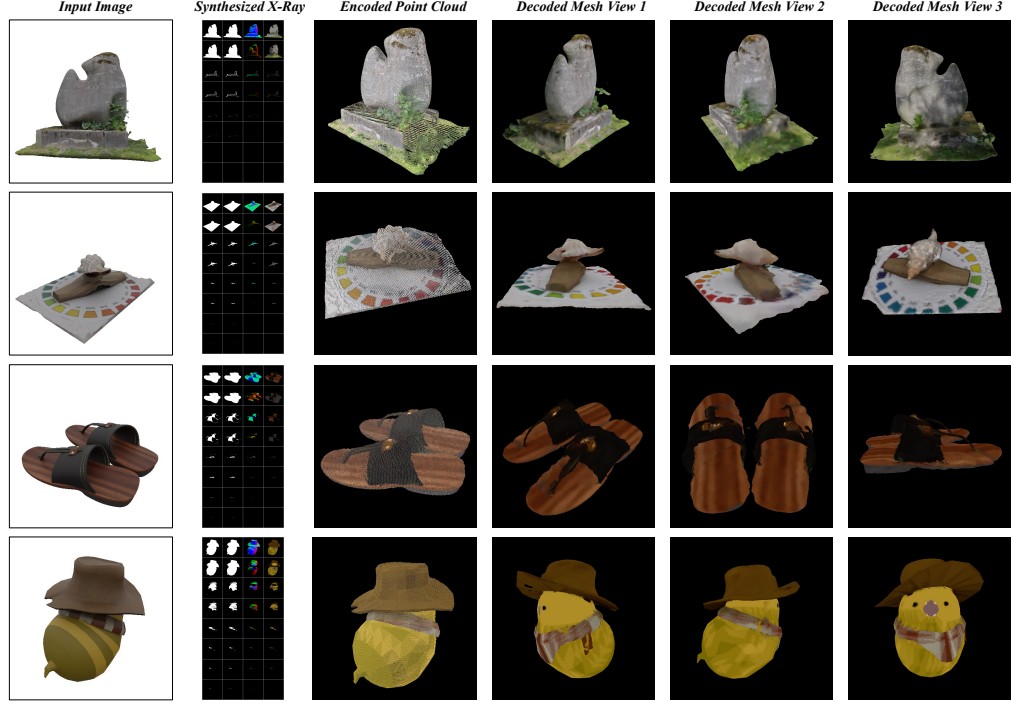

| Input Image | Synthesized X-Ray | Encoded Point Cloud | Decoded Mesh View 1 | Decoded Mesh View 2 | Decoded Mesh View 3 |

Figure 7: Visualization of Image-to-3D Generation from X-Ray.

model. These findings provide insights into the optimal configuration for diffusion models in 3D related tasks.

Table 3: Quantitative reconstruction comparison in different diffusion model configurations

| Model Configurations | CD ↓ | FS@0.1 ↑ | Training Time (days) ↓ | Inference Time (seconds) | Batch Size ↑ | Model Size (GB) ↓ |
|---|---|---|---|---|---|---|
| Finetuned Original UNet | 0.095 | 0.812 | ∼ 14 | ∼ 18 | 2 | 6.1 |
| Randomly Initialized Original UNet | 0.099 | 0.806 | ∼ 14 | ∼ 18 | 2 | 6.1 |
| Randomly Initialized UNet with 10% Parameters | **0.056** | **0.973** | **7** | ∼ 7 | **24** | **0.6** |

**The Effect of Hit H**

The original surface attributes only contain depth $\mathbf{D}$, normal $\mathbf{N}$, and color $\mathbf{C}$. We add an additional Hit $\mathbf{H}$ attribute to indicate whether there is a surface. In this ablation study, we demonstrate the necessity of including the Hit $\mathbf{H}$ attribute. We conduct training experiments with and without the Hit $\mathbf{H}$ attribute and evaluate on the GSO [6] dataset. The results, shown in Tab. 4, indicate that including the Hit $\mathbf{H}$ attribute can improve performance. The reason might be that the X-Ray is sparse and requires an indicator to balance the generative process. The UNet model and Upsampler with the Hit $\mathbf{H}$ attribute achieves better CD and FS@0.1 scores, demonstrating its importance in accurate 3D generation.

Table 4: Quantitative evaluation of the effect of Hit $\mathbf{H}$ attribute on the GSO [6] dataset

| Diffusion Model | CD ↓ | FS@0.1 ↑ | Upsampler | CD ↓ | FS@0.1 ↑ |
|---|---|---|---|---|---|
| wo. Hit $\mathbf{H}$ | 0.074 | 0.901 | wo. Hit $\mathbf{H}$ | 0.060 | 0.956 |
| w/ Hit $\mathbf{H}$ | **0.068** | **0.934** | w/ Hit $\mathbf{H}$ | **0.056** | **0.973** |

## A.4 More Visualization

We shown more Visualization results of single-view image generation in Fig. 7. Also, we extend the task to generate 3D object from text prompt. Text-to-3D mesh generation can also be achieved via using text-to-image, object segmentation, and image-to-3D processes. We utilize established diffusion models that are already proficient in image synthesis from textual descriptions instead of

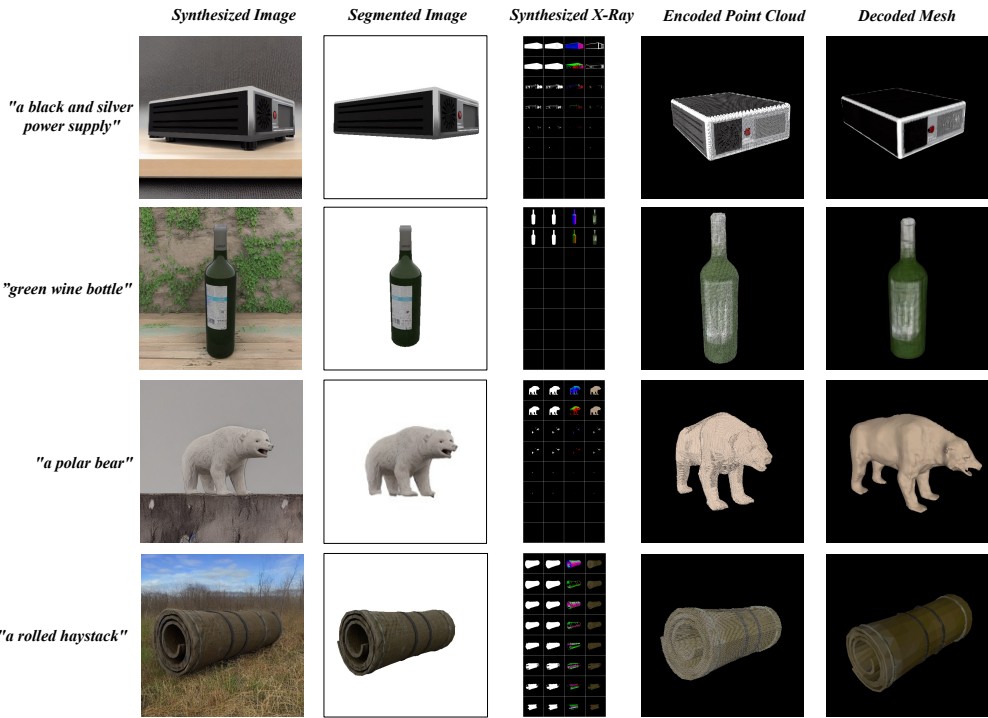

| Synthesized Image | Segmented Image | Synthesized X-Ray | Encoded Point Cloud | Decoded Mesh |

Figure 8: Visualization of Text-to-3D Generation from X-Ray.

developing a new text-conditioned diffusion model. One Model such as Stable Diffusion [43], Stable Cascaded [40], or DiT [39] are employed to generate images based on the input text. Following this, we apply an image segmentation tool, specifically the Segment Anything Model [22] to eliminate the background. This streamlined method avoids the complexities of training a new model from scratch instead of making use of sophisticated pre-trained models to handle the text-to-image translation, thereby simplifying the process of generating 3D meshes from textual inputs. The output results of Image-to-3D and Text-to-3D are illustrated in Fig. 8.

## A.5 Key Source Code

**Ray Casting:** The source code of ray casting the mesh to get the point cloud.

```
"""
The source code of ray casting the mesh to get the point cloud.
"""

from trimesh.ray.ray_pyembree import RayMeshIntersector

def ray_cast_mesh(mesh, rays_origins, ray_directions):
    intersector = RayMeshIntersector(mesh)
    index_triangles, index_ray, point_cloud = intersector.intersects_id(
        ray_origins=rays_origins,
        ray_directions=ray_directions,
        multiple_hits=True,
        return_locations=True)
    return index_triangles, index_ray, point_cloud
```

**X-Ray to Point Cloud:** The source code of transfering X-Ray to Point Clouds with normals and colors.

```
"""
The source code of transfering X-Ray to Point Clouds with normals and
    colors.
```

```python
"""

import numpy as np
import open3d as o3d

def get_rays(directions, c2w):
    # Rotate ray directions from camera coordinate to the world
    coordinate
    rays_d = directions @ c2w[:3, :3].T  # (H, W, 3)
    rays_d = rays_d / (np.linalg.norm(rays_d, axis=-1, keepdims=True)
    + 1e-8)

    # The origin of all rays is the camera origin in world coordinate
    rays_o = np.broadcast_to(c2w[:3, 3], rays_d.shape)  # (H, W, 3)

    return rays_o, rays_d

def X_Ray_to_Point_Cloud(XRay):
    """
    Converts X-Ray data to a point cloud with normals and colors.
    """
    XDepths, XNormals, XColors, XHits = XRay[:, 0:1], XRay[:, 1:4],
    XRay[:, 4:7], XRay[:, 7:8]

    camera_angle_x = 0.8575560450553894
    image_width = XDepths.shape[-1]
    image_height = XDepths.shape[-2]
    fx = 0.5 * image_width / np.tan(0.5 * camera_angle_x)

    rays_screen_coords = np.mgrid[0:image_height, 0:image_width].
    reshape(
        2, image_height * image_width).T  # [h, w, 2]

    grid = rays_screen_coords.reshape(image_height, image_width, 2)

    cx = image_width / 2.0
    cy = image_height / 2.0

    i, j = grid[..., 1], grid[..., 0]

    directions = np.stack([(i - cx) / fx, -(j - cy) / fx, -np.
    ones_like(i)], -1)  # (H, W, 3)

    c2w = np.eye(4).astype(np.float32)

    rays_origins, ray_directions = get_rays(directions, c2w)
    rays_origins = rays_origins[None].repeat(XDepths.shape[0], 0)
    ray_directions = ray_directions[None].repeat(XDepths.shape[0], 0)

    XDepths = XDepths.transpose(0, 2, 3, 1)
    XNormals = XNormals.transpose(0, 2, 3, 1)
    XColors = XColors.transpose(0, 2, 3, 1)

    rays_origins = rays_origins[XHits]
    ray_directions = ray_directions[XHits]
    XDepths = XDepths[XHits]
    normals = XNormals[XHits]
    colors = XColors[XHits]
    xyz = rays_origins + ray_directions * XDepths

    # convert to open3d point cloud
    xyz = xyz.reshape(-1, 3)
    normals = normals.reshape(-1, 3)
    colors = colors.reshape(-1, 3)
```

```
63    pcd = o3d.geometry.PointCloud()
64    pcd.points = o3d.utility.Vector3dVector(xyz)
65    pcd.normals = o3d.utility.Vector3dVector(normals)
66    pcd.colors = o3d.utility.Vector3dVector(colors)
67
68    return pcd
```

**Point Cloud to Mesh:** The source code of transferring the predicted point cloud to mesh using Poisson Surface Reconstruction.

```
1  """
2  The source code of transferring the predicted point cloud to mesh
       using
3  Poisson Surface Reconstruction.
4  """
5
6  import open3d as o3d
7
8  # Load point cloud
9  pcd = o3d.io.read_point_cloud("path_to_your_point_cloud.ply")
10
11 def poisson_surface_reconstruction(pcd):
12     """
13     Converts a point cloud to a mesh using Poisson Surface
       Reconstruction.
14     """
15     # Ensure the point cloud has normals
16     if not pcd.has_normals():
17         pcd.estimate_normals()
18
19     # Perform Screened Poisson Surface Reconstruction
20     mesh, densities = o3d.geometry.TriangleMesh.
       create_from_point_cloud_poisson(
21         pcd, depth=9, width=0, scale=1.1, linear_fit=False
22     )
23
24     # Optionally crop the mesh using the density values to remove low-
       density areas
25     # You can adjust the threshold based on your requirements
26     density_threshold = 0.01
27     vertices_to_remove = densities < density_threshold
28     mesh.remove_vertices_by_mask(vertices_to_remove)
29
30     # Assign colors to the mesh
31     mesh.vertex_colors = o3d.utility.Vector3dVector(pcd.colors)
32     return mesh
```

```
1  """
2  The source code of evaluating the predicted mesh using Chamfer
       Distance and F-Score.
3  """
4
5  import numpy as np
6  from scipy.spatial import cKDTree
7
8  def chamfer_distance_and_f_score(P, Q, threshold=0.1):
9      """
10     Calculates the Chamfer Distance and F-Score between two point
       clouds.
11     """
12     kdtree_P = cKDTree(P)
13     kdtree_Q = cKDTree(Q)
14
15     dist_P_to_Q, _ = kdtree_P.query(Q)
16     dist_Q_to_P, _ = kdtree_Q.query(P)
```

```
17
18      chamfer_dist = np.mean(dist_P_to_Q) + np.mean(dist_Q_to_P)
19
20      precision = np.mean(dist_P_to_Q < threshold)
21      recall = np.mean(dist_Q_to_P < threshold)
22
23      if precision + recall > 0:
24          f_score = 2 * (precision * recall) / (precision + recall)
25      else:
26          f_score = 0.0
27
28      return chamfer_dist, f_score
```

**Normalized Metrics:** The source code of normalize and align the predicted point cloud to the gt point cloud before evaluation.

```
1  """
2  The source code of normalize and align the predicted point cloud to
       the
3  gt point cloud before evaluation.
4  """
5  import numpy as np
6  import open3d as o3d
7
8  def ray_cast_mesh(mesh, rays_origins, ray_directions):
9      """
10     Performs ray casting on a mesh to obtain a point cloud.
11     """
12     intersector = RayMeshIntersector(mesh)
13     index_triangles, index_ray, point_cloud = intersector.
       intersects_id(
14         ray_origins=rays_origins,
15         ray_directions=ray_directions,
16         multiple_hits=True,
17         return_locations=True)
18     return index_triangles, index_ray, point_cloud
```

