# OpenReview forum: "X-Ray: A Sequential 3D Representation For Generation"
_NeurIPS.cc/2024/Conference — NeurIPS 2024 spotlight_

### Official Review · Reviewer_5c6P · 2024-07-08

**Soundness:** 3
**Presentation:** 3
**Contribution:** 3
**Rating:** 5
**Confidence:** 4

**Summary:**

The paper presents X-Ray, a new 3D sequential representation inspired by the penetrating quality of X-ray scans. This technique converts a 3D object into surface frames at different layers, ideal for creating 3D models from images. Experimental findings show that the X-Ray approach outperforms existing methods in improving the precision of 3D generation, opening up new possibilities for research and real-world applications in 3D representation.

**Strengths:**

* The motivation is interesting. Exising methods indeed cannot completely generate objects that include both visible and hidden surfaces. While the proposed method can the hidden interior of the object can be fully reconstructed.
* The compatibility of X-Ray data structures with sequential 3D representations in video formats opens up new opportunities for using video diffusion models in 3D generation.
* Great experimental results. The paper outperforms competitors by achieving the best results.

**Weaknesses:**

* The author asserts that "General, accurate, and efficient 3D representations are three crucial requirements for 3D generation." However, the citation is missing or the author should provide evidence to support the claim that 3D representations need to be general, accurate, and efficient.
* The ablation study is insufficient. The author should conduct ablation on X-Ray frames, for example, by using video or image diffusion techniquesm or the impact of frame complexity.
* The author claims to have synthesized a complete interior mesh in Figure 1, but I did not observe any other results besides cars or analysis in the experiments.
* I think the method used is the X-ray edge method because the dataset quality does not allow for a complete X-ray scan. For instance, the 3D CAT data has empty spaces within.

**Questions:**

* The authors need to provide a clear explanation of X-ray and complete internal mesh, as some training data contain empty spaces within.
* The authors should conduct additional ablation studies.

**Limitations:**

The authors acknowledged limitations but did not address the potential negative societal impact of the proposed technology.

---

> ### Author Rebuttal · Authors · 2024-08-06
>
> # To Reviewer 5c6P
> ### Question 1: General, accurate, and efficient 3D representations
> The author asserts that "General, accurate, and efficient 3D representations are three crucial requirements for 3D generation." However, the citation is missing or the author should provide evidence to support the claim that 3D representations need to be general, accurate, and efficient.
>
> ### Response 1
> We are grateful for the reviewer's feedback for our missing of citation. The statement "General, accurate, and efficient 3D representations are three crucial requirements for 3D generation" is based on the common understanding in the 3D generation community, and cited from [1].
> * General: A 3D representation should be able to represent a wide range of 3D objects, scenes, and shapes [2]. For example, voxel grids are general representations to any 3D shape, but they are computationally expensive.
> * Accurate: A 3D representation should accurately capture the geometry, appearance, and structure of 3D objects. For example, point clouds are accurate representations that capture the exact 3D points of an object, but they lack connectivity information [3].
> * Effcient: A 3D representation should be efficient in terms of memory usage, computational cost, and training time. For example, 3DGS [4] are efficient representations that can generate high-quality 3D shapes with low memory and computational cost.
>
> ### Question 2: Synthesizing a complete interior mesh
> The author claims to have synthesized a complete interior mesh in Figure 1, but I did not observe any other results besides cars or analysis in the experiments.
> ### Response 2
> We appologize for missing a detailed description to the concept of inside surface. More results in "Figure 5: Quantitative Comparison in Image-to-3D Generation" have illustrated the effectiveness of our method in synthesizing a __complete interior mesh__. Here are the explaination:
> * In the GSO dataset shown in Figure 5, our method is able to capture the inside surface of the shoe, while other methods fail to detect it and hide the welt;
> * Also, the OmniObject3D dataset in Figure 5 showcases our method's ability to capture the inside surface of objects such as cupboards and bowls. Other methods fail to accurately predict the inside surface under the input view.
>
> ### Question 3: Complete Internel Mesh Dataset.
> The authors need to provide a clear explanation of X-ray and complete internal mesh, as some training data contain empty spaces within.
> ### Response 3
> For datasets obtained through 3D scanning using multi-view images or depth sensors, it is true that the inside of the objects appears empty. However, the Objaverse dataset contains a variety of 3D models with and without interior meshes, which is why we conducted experiments on this dataset. Our X-Ray method outperforms the state-of-the-art method TripoSR by a large margin on this dataset. We acknowledge the reviewer's concern regarding the difficulty in obtaining datasets with interior details. However, as 3D modeling techniques advance, we anticipate that future datasets will include more detailed 3D models contains the inside mesh. Furthermore, we have found that indoor scenes and 3D CAD models are now accessible for training, providing abundant interior details that can be effectively learned by the X-Ray. As a result, we are able to synthesize complete scene-level 3D rooms or buildings with intricate interior details. We will include the results of these datasets in the revised paper.
>
> ### Question 4: Ablation studies
> The ablation study is insuffcient. The author should conduct ablation on X-Ray frames, for example, by using video or image diffusion techniquesm or the impact of frame complexity.
> ### Response 4
> * We appreciate the reviewer's suggestion for additional ablation studies. Due to limited computational resources, we were unable to conduct extensive ablation studies on arbitary X-Ray frame. However, previously, we first selected 16 X-Ray frames to represent each 3D object and then found that using only 8 frames achieved very close performance.
> * Additionally, we conducted an experiment to analyze the Encoding-Decoding Intrinsic Error, as shown in Figure 4 of paper. This is why we chose to use 8 frames in our experiments. The previous experimental results are as follows:
>
>     |Method|CD(L1) &darr;|FS@0.1 &uarr;|
>     |-|-|-|
>     |X-Ray w/ 16 frames|0.053|0.982|
>     |X-Ray w/ 8 frames|0.056|0.973|
>     |
>
> * Besides, we attempted to use the image diffusion model as our backbone for the lightweight computation. However, it did not produce satisfactory results, which is because the image diffusion model only considers the 2D spatial relation and cannot effectively capture the relationship between different frames. Consequently, we made the decision to switch to an off-the-shelf Stable Video Diffusion method for generating X-Ray frames. This method is better suited for addressing the sequential generation problem in 3D reconstruction. The previous experimental results are as follows:
>
>     |Backbone|CD(L1) &darr;|FS@0.1 &uarr;|
>     |-|-|-|
>     |Stable Diffusion|0.114|0.651|
>     |Stable Diffusion + Frame Attention|0.062|0.936|
>     |Stable Video Diffusion |__0.056__|__0.973__|
>     |
>
> ### Reference
> [1] Xiaoyu Li, Qi Zhang, Di Kang, Weihao Cheng, Yiming Gao, Jingbo Zhang, Zhihao Liang, Jing Liao, Yan-Pei Cao, Ying Shan. "Advances in 3D Generation: A Survey". arXiv:2401.17807.
>
> [2] Zhen Liu, Yao Feng, Yuliang Xiu, Weiyang Liu, Liam Paull, Michael J. Black, Bernhard Schölkopf. "Ghost on the Shell: An Expressive Representation of General 3D Shapes". ICLR2014.
>
> [3] Mescheder, Lars and Oechsle, Michael and Niemeyer, Michael and Nowozin, Sebastian and Geiger, Andreas. "Occupancy Networks: Learning 3D Reconstruction in Function Space". CVPR2019.
>
> [4] Kerbl, Bernhard and Kopanas, Georgios and Leimk{\"u}hler, Thomas and Drettakis, George. "3D Gaussian Splatting for Real-Time Radiance Field Rendering". ACM Transactions on Graphics.

---

### Official Review · Reviewer_wtWj · 2024-07-11

**Soundness:** 4
**Presentation:** 4
**Contribution:** 4
**Rating:** 9
**Confidence:** 5

**Summary:**

The paper addresses the problem of missing mesh interiors in image-to-3D generation. The proposed representation, X-Ray, adopts ray casting to encode both visible and hidden surfaces into a video format. With this representation, the authors enable single image-to-3D mesh generation, including the inside of the mesh, through a video diffusion model. Experiments demonstrate the effectiveness of the proposed method in 3D generation from a single image, both quantitatively and qualitatively.

**Strengths:**

(1) The task of single-view image to 3D mesh generation including interior surface of mesh is extremely challenging and well-motivated. As far as I know, this is the first paper that generates a mesh considering the interior surface.

(2) The idea of using ray casting to scan the interior of the mesh and convert it into video frames is novel. I really like this approach.

(3) The results show that the proposed method achieves SOTA performance on single image-to-3D generation and point cloud generation with a short inference time compared to existing baselines.

(4) The writing is easy to understand and the design choices are well presented.

**Weaknesses:**

(1) Considering Computed Tomography, it is expected that the performance will significantly increase as the number of layer $L$ increases. However, the experiments analyzed in Section 5.2 show that performance improvements are very minimal when $L$ is larger than 8. If additional computing resources are available, it would be interesting to see the performance when the frame resolution is low, but $L$ is extremely high (not mandatory).

(2) The entire training pipeline is quite long and memory-intensive. It was conducted on 8 NVIDIA A100 GPU servers for a week.

**Questions:**

(1) When training, it seems necessary to normalize the object so that the rays from a specific camera position can capture the entire object, making it easier to train. I'm curious whether this preprocessing step was applied to the Objaverse mesh. If this process was included, it would be beneficial to add this information to the paper.

(2) How much GPU memory is required during inference?

**Limitations:**

The authors mention the limitations related to the number of sequence layers and the missing parts when X-Ray is truncated, as discussed in Section 6 and Appendix A.5.

---

> ### Author Rebuttal · Authors · 2024-08-06
>
> # To Reviewer wtWj
> We appreciate the reviewer's highly positive rating to our paper! We address the reviewer's further concerns and questions in the following responses.
> ### Questions 1: Performance of Computed Tomography
> Considering Computed Tomography, it is expected that the performance will significantly increase as the number of layer L increases. However, the experiments analyzed in Section 5.2 show t performance improvements are very minimal when L is larger than 8. If additional computing resources are available, it would be interesting to see the performance when the frame resolution is but L is extremely high (not mandatory).
>
> ### Response 1
> We understand reviewer's concern about the performance when using all frames without omitting any surface. Indeed, conducting experiments with extremely high layers using Diffusion Models is not feasible for GPU memory limitation. However, this issue can be resolved via autoregressive Large-Large Model (LLM) to generate varying numbers of surfaces for each ray, ranging from 0 to even 100+, depending on the actual surface layers for each ray. With LLM, we can flexiblely model both indoor and outdoor scenes using X-Ray. We are going to release the excited research work in the future!
>
>
> ### Questions 2: Training Resources
> The entire training pipeline is quite long and memory-intensive. It was conducted on 8 NVIDIA A100 GPU servers for a week.
>
> ### Response 2
> We appreciate the reviewer's concern about the training pipeline. The training pipeline is relative long and memory-intensive, which is a common issue in training large-scale 3D models. For example, previous state-of-the-art method, Open-LRM, trained their model on 64 NVIDIA V100 GPUs for 5-6 days. Given our limited GPU resources, we made efforts to optimize and minimize the computational requirements as much as possible.
>
> ### Questions 3: Normalization
> When training, it seems necessary to normalize the object so that the rays from a specific camera position can capture the entire object, making it easier to train. I'm curious whether this preprocessing step was applied to the Objaverse mesh. If this process was included, it would be beneficial to add this information to the paper.
>
> ### Response 3
> The reviewer is correct. We did apply a preprocessing step to normalize the object, ensuring that the rays from a specific camera position can capture the entire object. The normalization code for arbitrary 3D objects is as follows:
>
> ```python
> def normalize_object():
>     bbox_min, bbox_max = scene_bbox()
>     scale = 1 / max(bbox_max - bbox_min)
>     for obj in scene_root_objects():
>         obj.scale = obj.scale * scale
>     # Apply scale to matrix_world.
>     bpy.context.view_layer.update()
>     bbox_min, bbox_max = scene_bbox()
>     offset = -(bbox_min + bbox_max) / 2
>     for obj in scene_root_objects():
>         obj.matrix_world.translation += offset
>     bpy.ops.object.select_all(action="DESELECT")
> ```
>
>
> ### Questions 4: GPU Memory
> How much GPU memory is required during inference?
>
> ### Response 4
> During inference, the GPU memory required is 4.8 GB for X-Ray diffusion model and 2.5 GB for X-Ray Upsampler. Thank you for the question, we will include this information in the revised paper.

---

> > ### Comment · Reviewer_wtWj · 2024-08-12
> > **Response to rebuttal**
> >
> > Thanks for the authors' response. I have read it as well as other reviews.
> >
> > I believe this paper presents a new problem (single image to mesh generation including interior surface) that the academic community has not yet solved, and I consider it the first paper to suggest a new direction for research. Additionally, the proposed method (using ray casting + video diffusion model) is novel and has been proven through experiments to be superior to existing methods, which I believe can serve as a good reference for future research.
> >
> > Regarding the concern about the number of $L$, which I and other reviewers were worried about, the authors claimed to address it through an autoregressive Large-Language Model (LLM), but this part is not explained in detail, so it's hard to understand. However, given the current limitations in computing resources, the performance is sufficiently guaranteed even with $L=8$, to the extent that it surpasses other baselines, so I don’t think this is a major issue.
> >
> > In the revised version, it would be good to make revisions to address the concerns of the other reviewers, and particularly, I strongly recommend adding a section on the efficiency of the X-Ray representation (Rebuttal to Reviewer xHzv, Response 2). Since a new representation has been proposed, if a comparison with other representations is organized in a table, readers will be able to easily grasp the efficiency of the X-Ray representation at a glance.
> >
> > I appreciate the additional experiments conducted for the rebuttal, and for the reasons mentioned above, I will maintain my previous rating of 9.

---

> > > ### Author Response · Authors · 2024-08-12
> > >
> > > Thank you once again for recognizing the value of our work and maintaining the rating of 9, which is highly encouraging for us and our future efforts to advance X-Ray representation in the 3D domain! We have carefully considered the reviewers’ comments and will add an experimental section on the efficiency of the X-Ray representation to enhance clarity. As mentioned in our response to Reviewer xHzv, we compare it with other 3D representations, such as Voxels, point clouds, and multi-view depths, to make it easier for readers to grasp its efficiency.

---

### Official Review · Reviewer_EY92 · 2024-07-11

**Soundness:** 3
**Presentation:** 3
**Contribution:** 3
**Rating:** 5
**Confidence:** 4

**Summary:**

The paper proposes a new representation that encodes a 3D mesh into the ray intersection points from a single view point.  The position, color, and surface normal at the intersection points are stored as the representation. Poisson reconstruction is used to recover the 3D mesh from the representation. A cascaded diffusion model is trained using the extracted representation from 60k objaverse meshes. Evaluation is conducted on three other datasets.

**Strengths:**

The paper proposes a new representation, which is compact, efficient, and tailored to single image to 3D applications. It demonstrated better performance compared to other methods in the paper in the metrics. The visual quality of the results are also better.  3D generation is a very relevant topics and the paper will be interesting to many readers.

**Weaknesses:**

1. The paper does not discuss how to extend the proposed representation to include multiple viewpoints to provide better encoding/decoding quality. Even though the representation records the intersection points along a ray, the intersection points from front and top views would be very different.

2. One limitation of X-ray is that the limited number of intersection points (8 in the paper) makes it difficult to encode complex shapes like hairs and carefully designed meshes with all internal structures. For example, taking a real engine, the number of surfaces a ray can pass through would be far more than 8 (eg., the wires, the bores, shafts).  The paper mentions the limitation, however, it does not seem clear to me from the paper how the limitation can be addressed efficiently.

3. Another related limitation is the potentially nonuniform number of intersection points along different rays from the same viewpoint. For example, on a human head, the rays passing through the hair region would need a much higher number of intersections to avoid losing information than those through the face region.  The nonuniformity extends to across objects.  To model one object that has a large number of intersection points, it might need to increase L used for all objects in order to learn a diffusion model.  This property makes the presentation inefficient.

4. In order to generate all interior structures of a 3D object, we need meshes that are carefully designed with interior details.  This kind of dataset is difficult to get and if existed would increase L, making the method inefficient.

**Questions:**

1. The proposed representation is related to the pointmap used in DUSt3R (CVPR2024).  From my understanding, at a high level, it extends pointmaps to multiple points along a ray. To me, it is an interesting connection and can be mentioned in the paper.

2. The core of the representation is finding the intersection points and creating the tensor representation. Currently the paper only mentions that it calls a function in trimesh.  I would suggest describing the operation in more detail -- for example, I do not know how the Hit is constructed. Does the method try to aggregate intersection points at similar depths at the same index along the L-dimension?  Or is Hit simply a padding mechanism?

3. In Sec 5.3 quantitative comparison, Table 1 and Table 2 show metrics that evaluate the quality of the geometry. How is the photometric quality of the proposed method compared to other methods, e.g., PSNR or SSIM on the conditioning viewpoint (and other novel views)?

4. How significant is the effect of the predicted normal, and is it fair to compare rendering-based methods like LRM using the extracted meshes in Table 1?  My understanding is the quality of the Poisson reconstruction can be affected heavily by the quality of the normal, which may be poorly modeled by volumetric density.  Should the paper also show the same metrics directly on the point clouds or depth maps?

**Limitations:**

The paper mentions the limited number of intersection points. However, it does not seem clear how to address the limitation potentially without significantly increasing the number of L.

---

> ### Author Rebuttal · Authors · 2024-08-06
>
> # To Reviewer EY92
> ### Question 1: Multiple viewpoints Extension.
> How to extend the proposed representation to include multiple viewpoints to provide better encoding/decoding quality.
> ### Response 1
> we aim to show the advantage of the X-Ray by providing a simple baseline for single-view 3D reconstruction. While extending this to multiple viewpoints is interesting, it might not be necessary for this task. To address the reviewer’s concern, we encoded 1000 randomly selected 3D meshes into X-Ray from different viewpoints, like front and top views, and then decoded them back into 3D meshes. The standard deviation of reconstruction error from different views is negligible (CD < 1e-3). This demonstrates the robustness of the X-Ray to multiple viewpoints and its ability to reconstruct 3D meshes from any viewpoint. We will include this discussion in the revised paper.
>
> ### Question 2: Limitation of finite and nonuniform intersection points
> One limitation of X-ray is that the limited number of intersection points makes it difficult to encode complex shapes like hairs. Another limitation is nonuniform number of intersection points makes the presentation inefficient.
> ### Response 2
> * In the experimental results (Figure 4 (a)/(b)), using 8 or even 6 frames is sufficient to cover most of the surfaces, as most 3D models are not highly detailed. However, for complex shapes such as hairs, this limitation can be addressed by increasing the number of frames, to encode more intricate details.
> * To further overcome the nonuniform number of surface layers limitation, autoregressive Large-Language Model (LLM) can be employed as the generator to handle surfaces with a dynamic number of layers. This approach allows for rays to generate varying numbers of surfaces, ranging from 0 to 100+, depending on the actual surface layers for each ray.
>
> ### Question 3: Dataset
> Carefully designed dataset is difficult to get and if existed would increase L making the method inefficient.
> ### Response 3
> Obtaining a dataset with interior details is a challenge, as mentioned by the reviewer. However, we are optimistic about the increasing availability of fine-grained 3D datasets. With advancements in 3D modeling, more detailed 3D models will likely be included in future datasets. Additionally, we have access to indoor scenes and CAD models, which contain rich interior details that can be effectively learned by the X-Ray. This allows us to synthesize complete scene-level 3D rooms or buildings with intricate interior details. As described above, LLM is our next version of generator to handle surfaces with a dynamic number of layers, which will be more efficient in generating complex shapes.
>
> ### Questions 4: Related Work: DUSt3R
> It is a interesting to connect and mention DUSt3R (CVPR2024).
> ### Response 4
> We will mention the connection to the work of DUSt3R in our revised paper as it has made significant contributions to multi-view depth and pose estimation.
>
> ### Question 5: Representation Operation
> Describe the representation operation in more detail.
> ### Response 5
> We have described the process of obtaining X-Ray from a 3D mesh in the Appendix PDF and source code. The "hit" value is determined by checking if the depth is greater than zero. We first identify the intersected mesh face index and then query its properties as X-Ray. Below is the pseudocode we provided:
>
> ```python
> from trimesh.ray.ray_pyembree import RayMeshIntersector
>
> def ray_cast_mesh(mesh, ro, rd):
>     intersector = RayMeshIntersector(mesh)
>     index_faces, _, _ = intersector.intersects_id(
>         ray_origins=ro, rd=rd,  multiple_hits=True)
>     return index_faces
>
> def Mesh2XRay(mesh, Rt, K):
>     # get camera center and ray direction
>     ros, rds = get_camera_ray(Rt, K)
>
>     XRay = []
>     for ro, rd in zip(ros, rds):
>         index_faces, = ray_cast_mesh(mesh, ro, rd)
>         depth, normal, color = extract_features(mesh, index_faces)
>         # calc hit
>         hit = depth > 0
>         xray = torch.cat([hit, depth, normal, color], dim=-1)
>         XRay += [xray]
>     XRay = torch.stack(XRay) # (H, W, L, 8)
>     return XRay
> ```
>
> ### Question 6: Photo-metric Quality
> How is the photometric quality of the proposed method compared to other methods, e.g., PSNR or SSIM on the conditioning viewpoint (and other novel views)?
> ### Response 6
> Novel view synthesis and single-view reconstruction are two distinct tasks. Evaluating the quality of 3D reconstruction based on photometric metrics may not be appropriate. For example, methods like NeRF and 3DGS can achieve high scores in novel-view synthesis at the photometric level, however, their extracted 3D shapes and textures often suffer from really poor quality. Ours focus is on accurately reconstructing the 3D ground-truth shape as the evaluation. We appreciate the reviewer's observation regarding photometric quality. We will predict additional 3D Gaussian Splatting parameters to synthesize high-quality novel view images in the revised paper.
>
> ### Question 7: Normal Prediction
> How significant is the effective of the predicted normal?
> ### Response 7
> Point normals can be estimated from the point cloud, but determining their orientation (inside or outside the surface) is challenging. To solve this, we generate additional point cloud normals for consistent orientation. We tested the importance of normal prediction for rendering methods like LRM and our method. By fine-tuning OpenLRM with normal supervision, we found that adding ground-truth normals did not significantly improve its reconstruction metric and is still not comparible with ours. However, normal prediction is crucial for our method due to Poisson surface reconstruction. This ablation study will be included in the revised paper.
>
> | Method       | Chanfer Distance &darr; |
> |--------------|----------|
> | OpenLRM w/o normals | 0.143     |
> | OpenLRM w/ normals | 0.138 (+3.5%)     |
> |
> | X-Ray w/o normals  | 0.067    |
> | X-Ray w/ normals  | 0.056  (+16.4%)   |
> |

---

> > ### Comment · Reviewer_EY92 · 2024-08-11
> >
> > I thank the authors for the reply. From Response 1, does it mean that it is difficult to extend x-ray to few-view to 3d use cases? Though x-ray can represent simple meshes well, the constraint of single-view only seems an important characteristic of the proposed method that should be discussed in the paper. For example, few-view (eg, 2-4) to 3d is also a frequently studied problem and common in practice.
> >
> > For Response 6, I understand and agree with the description about the difference between single-view to 3d and multiview reconstruction. However, the PSNR value on the given single view is still an informative and meaningful metric even for single view to 3d task.

---

> > > ### Author Response · Authors · 2024-08-11
> > > **Response to Reviewer EY92**
> > >
> > > ### We sincerely appreciate the reviewer's comments and the subsequent discussion. We will do our best to address the concerns raised.
> > >
> > > ### 1. Does it mean that it is difficult to extend x-ray to few-view to 3d use cases?
> > > * Although single-view X-Ray can solve most cases as a simple baseline, the reviewer suggests further strengthening the performance by introducing a multi-view solution.
> > >
> > > * Similar to multi-view depth representation, it is easy to extend X-Ray to a multi-view style by casting rays under multiple cameras then concatenating these frames as video. There will be $N \times L$ frames in the multi-view X-Rays, where $N$ is the number of views and $L$ is the number of surface layers in each view. We can then generate the multi-view X-Rays from a single image via the same video diffusion model.
> > >
> > > * We greatly appreciate the reviewer's suggestion and will focus on validating how multi-view X-Rays outperform single-view X-Ray generation in our subsequent research work.
> > >
> > >
> > > ### 2. PSNR value on the given single view is still an informative and meaningful metric even for single view to 3d task
> > > * The reviewer is very expert in novel view synthesis and 3D reconstruction and generation research. We thank the reviewer's comment that the PSNR value is an informative and meaningful metric for the single-view to 3D task.
> > >
> > > * For evaluation, the deeper question is which should be considered the ground truth for photometric performance: albedo (original surface color) or image (rendered color under lighting). We predict the albedo before rendering, while rendering-based methods predict the final image color after rendering. These are two different colors. Due to this ambiguity, we did not report these results. However, in our previous experiment using albedo as the ground truth, our X-Ray achieved 21.3 PSNR, whereas the state-of-the-art TripoSR only achieved 17.6 PSNR on the GSO dataset.
> > >
> > > * We understand that the reviewer might be referring to the performance of evaluating rendered colors, similar to other rendering-based methods. Our solution is to predict the 3DGS parameters for each surface of X-Ray and render the image to compare with the ground truth. As described in Response 1, as the first work, we aim to provide a simple baseline X-Ray representation and avoid involving more technical complexities. We will report both photometric performance results in the revised paper. Similar to the normalized shape evaluation in this paper, we hope the photometric evaluation under both albedo and image can provide a new benchmark for the community.

---

### Official Review · Reviewer_xHzv · 2024-07-12

**Soundness:** 3
**Presentation:** 3
**Contribution:** 3
**Rating:** 5
**Confidence:** 4

**Summary:**

This paper introduces X-Ray, a new 3D representation designed for efficient generation of 3D objects from single images. The key idea is to represent a 3D object as a sequence of 2D "surface frames" capturing hit/miss, depth, normal, and color information along rays cast from the camera viewpoint. This sequential representation lends itself well to generation using video diffusion models, enabling the synthesis of both visible and hidden surfaces. The authors propose a pipeline consisting of an X-Ray diffusion model to generate low-resolution surface frames from an input image, followed by an X-Ray upsampler to enhance resolution. They evaluate their method on single-view 3D reconstruction and unconditional 3D shape generation tasks, reporting quantitative results on standard benchmarks.

**Strengths:**

- The paper tackles the important challenge of reconstructing complete 3D models, including hidden surfaces, from single images. This is a significant limitation of current rendering-based approaches, and the authors' focus on this problem is well-motivated and timely.

-  The X-Ray representation, while its novelty requires further substantiation, offers an intuitive way to encode 3D surface information in a sequential manner. By focusing solely on surface details rather than volumetric data, the representation has the potential to be more memory-efficient than voxel grids or dense point clouds, especially for objects with complex internal structures.

- The authors smartly leverage recent advancements in video diffusion models for 3D generation. This is a promising direction, as video diffusion models have shown impressive capabilities in synthesizing high-quality and temporally coherent sequences of images. Adapting these models to the task of 3D generation through the X-Ray representation is a reasonable and potentially fruitful approach.

**Weaknesses:**

- The paper does not provide a convincing argument for the novelty of the X-Ray representation. While its sequential capture of surface information is intuitive, a thorough comparison to existing techniques is lacking. Specifically, the authors should clearly differentiate X-Ray from methods like depth peeling, multi-view depth images, multi-plane images (MPI), and notably, the PI3D representation (Liu et al., CVPR 2024), which also leverages diffusion models for 3D generation.
  * Liu, Ying-Tian, Yuan-Chen Guo, Guan Luo, Heyi Sun, Wei Yin, and Song-Hai Zhang. "Pi3d: Efficient text-to-3d generation with pseudo-image diffusion." In Proceedings of the IEEE/CVF Conference on Computer Vision and Pattern Recognition, pp. 19915-19924. 2024.

- The paper repeatedly emphasizes the efficiency of the X-Ray representation without providing concrete evidence or analysis. The authors should quantify their claims by comparing the memory footprint and computational costs of X-Ray to alternative representations like voxel grids, point clouds, and neural implicit representations (e.g., NeRFs) for objects of varying complexity.

- While leveraging video diffusion models is promising, the paper does not clearly articulate how the specific properties of the X-Ray representation are exploited within the diffusion process beyond being a sequential data format. Do the hit/miss indicators or the ordered nature of surface frames influence the model architecture or training? Would similar performance be achieved with alternative sequential representations as input to the diffusion model?

- The evaluation heavily relies on reconstruction metrics (CD, EMD), even when assessing a generative model. While these metrics are relevant for the single-view reconstruction task, they do not capture the generative capabilities of X-Ray. The authors should expand generative evaluation to diverse categories beyond ShapeNet Cars. The paper could assess generative quality by evaluating the diversity and realism of multiple shapes generated from the same input image.

- The evaluation would be also strengthened by: (1) Including recent SDS-based single-view 3D generation methods as baselines; (2) Providing visualizations of generated shapes for the unconditional generation experiment; (3) Dedicating a section to analyze failure cases, visually showcasing problematic inputs and outputs.

**Questions:**

- Could the authors please elaborate on the key differences between the X-Ray representation and existing techniques like depth peeling, multi-view depth images, and multi-plane images (MPI)? Also please include the discussion with PI3D.

- To support the claims of efficiency, could the authors provide a quantitative analysis of the memory footprint and computational cost (encoding, decoding, generation) of X-Ray compared to voxel grids, point clouds, or NeRFs? This analysis should consider objects of varying complexity and resolutions.

- Could the authors please include generative metrics for the single-view 3D reconstruction experiments on GSO and OmniObject3D?

**Limitations:**

- The authors merely state limitations without explaining their causes, impact, or potential solutions. E.g., saying "X-Ray frames become sparse" is not enough. How does this sparsity affect generation? How can it be addressed?

---

> ### Author Rebuttal · Authors · 2024-08-04
>
> # To Reviewer xHzv
> ### Question 1: Comparison to Existing Techniques
> A thorough comparison to existing techniques like depth peeling, multi-view depth images, MPI, and the PI3D.
> ### Response 1
> * We discussed multi-view images and MPI in the related work. Furthermore, Multi-view depth cannot sense the inside surface and might records the redundancy surfaces into nearby views. MPI divides the object into planes with fixed distances, while our X-Ray stores a dynamic number of surfaces. Refer to the following Table 1 for efficient comparison.
> * We apologize for not being familiar with depth peeling before submission. Our X-Ray and depth peeling serve different purposes. Depth peeling is mainly for rendering transparent surfaces, while X-Ray transform any 3D object in video format. Besides, our main contribution is using video diffusion as generator to generate objects.
> * The PI3D is an interesting approach that uses diffusion models for 3D generation. Since we do not have access to source code, to ensure a fair comparison, we will reimplement the PI3D method and include it in the revised paper.
>
> ### Question 2: Efficiency of the X-Ray
> The paper should provide more evidence and analysis to support its claims about the efficiency of the X-Ray.
> ### Response 2
> We compared the efficiency of different representations using 500 3D meshes from ShapeNet dataset. The results showed that both point cloud and X-Ray were highly efficient, with lower memory, faster encoding & decoding times. However, the X-Ray had the advantage of being reorganizable as a video format for diffusion models, leading to better performance.
> |Method|Memory (&darr;)|Encoding Method|Encoding Time (&darr;)|Decoding Method|Decoding Time (&darr;) |Reconstruction Error(CD)(&darr;)|Generation Metric (Cov-EMD) (&uarr;)|
> |-|-|-|-|-|-|-|-|
> |3D Grid|67.09 MB|Voxlization|0.105 s|Poisson|~5 s|7.7e-3|3D-DiT[36] &rarr; 56.38|
> |Multi-View Depths (8 views)|1.57 MB|Rendering via Blender|0.045 s|Fusion & Poisson|~10 s|1.1e-2|-|
> |MPI (8 planes)|1.57 MB|Slicing & Rendering via Blender|0.049 s|Poisson|~5 s|8.9e-3|-|
> |Point Cloud (200,000 points)|__0.90 MB__|Surface Sampling|__0.013__ s|Poisson|~5 s|__7.2e-3__|LION[59] &rarr; 56.53|
> |
> |X-Ray (8 layers)|__0.82 MB__|Ray Casting|__0.016 s__|Poisson|~5 s|7.8e-03|Ours &rarr; __60.27__|
> |
>
> Table 1. Comparison with other 3D representations in Efficiency.
>
> ### Question 3: Ablation Studies
> The paper should explain how the unique properties of the X-Ray are utilized in the diffusion process and whether other sequential representations could achieve similar performance.
> ### Response 3
> In section A.3 of the Appendix, we have conducted ablation studies to examine various aspects of the X-Ray model, including the impact of the hit/miss indicators. After exploring different formats, we transitioned to an off-the-shelf Stable Video Diffusion method, which proved to be more effective in addressing the sequential generation problem in 3D reconstruction.
>
> ### Question 4: Evaluation Metrics
> The evaluation should include a broader range of categories to assess the generative capabilities of X-Ray. Could the authors please include generative metrics for the single-view 3D reconstruction experiments on GSO and OmniObject3D?
> ### Response 4
> We chose to focus on cars as they have complex internal and external 3D surfaces, making them an ideal category to showcase the benefits of our method. Objaverse has over 1000 categories. Previous state-of-the-art methods only used reconstruction metrics for evaluation. Thank you for reminding us. We included generative metrics for the single-view 3D reconstruction experiments on GSO in the table below. Since our method can sense the inner surfaces, we have achieved overwhelming superiority. Any additional metrics will be included in the revised paper.
>
> |Method| 1-NNA (EMD) &darr; |Cov (EMD) &uarr;|
> |-|-|-|
> |One-2-3-45[24]|64.37|25.85|
> |OpenLRM[13]|59.14|38.31|
> |TripoSR[48]|57.25|40.69|
> |X-Ray|__52.42__|__48.27__|
> |
> Table 2. Generative metrics on GSO Dataset.
>
> ### Question 5: Additional Suggestions
> The evaluation would be also strengthened by: (1) Including recent SDS-based single-view 3D generation methods as baselines; (2) Providing visualizations of generated shapes for the unconditional generation experiment; (3) Dedicating a section to analyze failure cases, visually showcasing problematic inputs and outputs.
> ### Response 5
> 1. SDS-based methods like Zero-1-to-3 and DreamGaussian rely on diffusion loss to optimize NeRF or 3D Gaussian splatting. However, these methods are time-consuming, taking several minutes to generate a single 3D mesh from an image. This makes them impractical for evaluation on large datasets like GSO and OmniObject3D. They all did not report their 3D reconstruction metrics. In contrast, all rendering-based methods and our X-Ray can generate 3D meshes from images in just a few seconds. Additionally, SDS-based methods also struggle to generate inner surfaces, limiting their performance on datasets of GSO and OmniObject3D.
>
> 2. We have add more visualizations in the Appendix (Sec A.4). Specifically, Figures 6 and 7 showcase the arbitary 3D objects generated by our proposed method. We appreciate your feedback and are planning to further enhance the performance on the 3D CAD model dataset (abundant inside and outside meshes) in the revised paper.
>
> 3. We have included a dedicated section in the Appendix (Sec A.5) and Figure 8 to analyze failure cases, and we will enhance this section in the revised paper.
>
> ### Question 6: Solution to Limitation
> How does this sparsity affect generation? How can it be addressed?
> ### Response 6
> To further overcome the sparse related limitation, an autoregressive Large-Language Model (LLM) can be employed as the generator, allowing for each ray to generate varying numbers of surfaces, ranging from 0 to 100+, depending on the actual surface layers.

---

### Author Rebuttal · Authors · 2024-08-07

### Responses to All Reviewers:
We would like to express our sincere gratitude for the valuable feedback provided by the reviewers. It is truly encouraging to see that our X-Ray work has been given positive evaluation by all the reviewers. We appreciate the comments and constructive suggestions given by the reviewers.
Although more visualizations and ablation studies have been included in the Appendix of the submitted paper, we apologize that there are some confusions caused by unclear expressions, potentially inadequate comparisons with existing methods, and a lack of detailed explanations in the paper. In response, we have diligently addressed all the concerns and suggestions raised by the reviewers in the corresponding rebuttals.

---

### Decision · Program_Chairs · 2024-09-25

**Decision:**

Accept (spotlight)

**Comment:**

The paper received all positive recommendations. The area chair agreed with this recommendation.